# Genetic Associations of *ACOX2* Gene with Milk Yield and Composition Traits in Chinese Holstein Cows

**DOI:** 10.3390/ani15070953

**Published:** 2025-03-26

**Authors:** Hui Cao, Zhe Wang, Lingna Xu, Bo Han, Dongxiao Sun

**Affiliations:** State Key Laboratory of Animal Biotech Breeding, Key Laboratory of Animal Genetics, National Engineering Laboratory of Animal Breeding, Department of Animal Genetics and Breeding, College of Animal Science and Technology, China Agricultural University, Breeding and Reproduction of Ministry of Agriculture and Rural Affairs, Beijing 100193, China; 17334884025@163.com (H.C.); wz123@cau.edu.cn (Z.W.); xulingna@caas.cn (L.X.); bohan@cau.edu.cn (B.H.)

**Keywords:** genetic effect, *ACOX2*, milk traits, SNP, association analysis, dairy cattle

## Abstract

This study investigated the genetic effects of the acyl-CoA oxidase 2 (*ACOX2*) gene on milk production traits in Chinese Holstein cows. We identified five single nucleotide polymorphisms (SNPs) that significantly influenced 305-day milk, fat, and protein yields. In addition, we found that rs109066086 may regulate the transcriptional activity of the *ACOX2* gene. Overall, our research highlights the genetic effects of the *ACOX2* gene on milk yield and composition traits in dairy cattle and provides valuable molecular markers for genomic selection.

## 1. Introduction

Milk is an essential source of nutrients in the human diet, being rich in carbohydrates, protein, vitamins, and minerals [1]. Numerous studies have highlighted the biological functions of milk, such as its ability to lower blood pressure and enhance immune function, which has led to a higher demand for high-quality milk [1,2,3]. Improving both milk yield and quality has always been the primary goal of dairy cattle breeding.

Before the advent of genomic selection (GS), progeny testing was the primary method used in dairy cattle breeding. Although this method is highly accurate, it has a long cycle, typically 5 to 6 years [4]. GS has reduced the dairy breeding cycle to approximately two years, improved breeding accuracy, and reduced breeding costs by 92% [5]. Studies have shown that incorporating functional gene information into genomic-selection chips can improve the accuracy of genomic estimation of breeding values [6,7,8]. However, milk yield and composition traits are controlled by polygenes with minor effects [5,9]. So far, only diacylglycerol O-acyltransferase 1 (*DGAT1*), growth hormone receptor (*GHR*), and ATP-binding cassette subfamily G member 2 (*ABCG2*) genes have been confirmed as causal genes for milk production traits [10,11,12,13]. The development of sequencing technologies and multi-omics approaches has made it easier to identify functional genes associated with milk production traits in dairy cattle.

In our previous studies, proteomic analyses revealed that the expression of the ACOX2 was significantly higher in the livers of Holstein cows during the peak of lactation compared to the dry period (fold change ≥ 1.5, *p* = 6.75 × 10^−3^) [14]. The *ACOX2* gene in bovines is located on chromosome 22 and has 14 exons with a gene length of 30,884 bp. It is involved in fatty acids biosynthesis by oxidizing long-chain fatty acids to short-chain fatty acids in the peroxisome and catalyzing the conversion of cholesterol to bile acids (BAs) by oxidizing CoA ester intermediates of BAs. *ACOX2* deficiency leads to the accumulation of BAs and intermediates [15]. BAs can act as signaling molecules to regulate lipid, glucose, and energy metabolisms, and they promote the absorption of fat and fat-soluble vitamins from food [16,17]. Previous studies have found that increasing ACOX2 levels helps eliminate Staphylococcus aureus in bovine mammary cells [18]. In addition, ACOX2 is also associated with the metabolism of subcutaneous fat in beef cattle [19]. Of note, the *ACOX2* gene is located 3.14 cM from the QTL (ID: 2487) for milk protein percentage in dairy cattle [20]. Moreover, *ACOX2* is close to the SNP ARS-BFGL-BAC-33314, which is associated with milk fat and protein traits, with 1.8 Mb, and the SNP ARS-BFGL-NGS-61008, which is associated with the milk protein percentage, with 0.2 Mb [21]. The kyoto encyclopedia of genes and genomes (KEGG) pathway analysis (https://www.kegg.jp/kegg/pathway.html, accessed on 12 January 2025) also indicated that this gene is involved in primary bile acid biosynthesis, metabolic pathways, and the PPAR signaling pathway. Therefore, the *ACOX2* gene is regarded as a promising candidate for milk traits.

Therefore, the purpose of this study was to systematically evaluate whether the *ACOX2* gene has significant genetic effects on milk yield and composition traits, thereby providing valuable genetic markers for genomic selection programs in dairy cattle.

## 2. Materials and Methods

### 2.1. Animals and Phenotypic Data Collection

The experimental population comprised 922 Chinese Holstein cows from 44 sire families. These cows were collected from the 21 dairy farms of Beijing Sunlon Livestock Development Co., Ltd. (Beijing, China) and were raised under uniform feeding conditions; their pedigree information and phenotypic data (including 305-day milk yield, milk fat yield, milk fat percentage, milk protein yield, and milk protein percentage) were provided by the Beijing Dairy Cattle Center (Beijing, China). Descriptive statistics of the phenotypic values for milk production traits across the two lactations are presented in Appendix A.

### 2.2. Genomic DNA Extraction

Genomic DNA was extracted from frozen semen of 44 Holstein bulls using the optimized high-salt method and from whole blood samples of 922 daughters using the TIANamp Blood DNA Kit (Tiangen, Beijing, China). The quantity and quality of the extracted DNA was assessed using a NanoDrop 2000 Spectrophotometer (Thermo Scientific, Hudson, NH, USA) and 2% agarose gel electrophoresis, respectively.

### 2.3. SNP Identification and Genotyping

Using Primer 3.0 (https://primer3.ut.ee/, accessed on 20 May 2024), 21 pairs of primers were designed (Appendix A) to amplify the entire coding region and 2000 bp of the 5′ and 3′ flanking regions for the *ACOX2* gene (Gene ID: 514969) and were synthesized at the Beijing Genomics Institute (BGI, Beijing, China). Genomic DNA from 44 bulls was diluted to a final concentration of 50 ng/μL. The 44 DNA samples were randomly divided into two DNA pools with 22 samples each. We performed the PCR for the amplifications (Appendix A). PCR products were detected via 2% agarose gel electrophoresis, and the qualified products were sequenced at Beijing Qingke Xinye Biotechnology Co., Ltd. (Beijing, China). Chromas software 2.6 was used to check the sequence diagram and compare it with the bovine reference genome sequence (ARS-UCD1.2) to identify polymorphic sites. Genotypes of individual SNPs of 922 cows were identified using the Genotyping by Target Sequencing (GBTS) method at Shijiazhuang MolBreeding Biotechnologu Co., Ltd. (Shijiazhuang, China).

### 2.4. Linkage Disequilibrium (LD) Estimation

We estimated the linkage disequilibrium (LD) between the identified SNPs using Haploview 4.2 (Broad Institute of MIT and Harvard, Cambridge, MA, USA). The magnitude of LD is represented by the D’ value, and haplotypes with frequencies greater than 0.05 were retained.

### 2.5. Association Analyses

We performed association analyses between the SNPs and/or haplotypes and five milk traits using the mixed procedure of SAS 9.4 software (SAS Institute Inc., Cary, NC, USA) with the following animal model: y=μ+HYS+b×M+G+a+e
where y represents the phenotypic value of each trait for each cow; μ is the population mean; HYS includes the fixed effects of farm (1–21: 21 farms), calving year (1–4: 2012–2015), and season (1: April–May; 2: June–August; 3: September–November; 4: December–March); M is the month of calving as a covariant, where b is its regression coefficient; G is the effect of the genotype or haplotype combination; a is the individual random additive genetic effect, distributed as $N(0,A\delta_{a}^{2})$, with the additive genetic variance δ_a^2; and e is the random residual, distributed as $N\left(0,I\delta_{e}^{2}\right)$, with the identity matrix I and the residual error variance $\delta_{e}^{2}$. 

Also, we calculated the additive effect (a), dominant effect (d), and substitution effect (α) using the formulas:$$a=\frac{AA-BB}{2};d=AB-\frac{AA+BB}{2};\;\alpha =a+d(q-p)$$
where AA, BB, and AB are the least squares means of each trait for the respective genotypes, p is the frequency of allele A, and q is the frequency of allele B.

The genetic effect of an SNP on a particular trait can be expressed by the proportion of phenotypic variation it explains. The formula for calculating the phenotypic variance explained by an SNP is as follows:$$Phenotypic\;variance\;ratio\;(PVR)=\frac{2pq\alpha^{2}}{\sigma_{p}^{2}}$$
where p and q are the allele frequencies for allele A and allele B, α is the substitution effect, and $\sigma_{p}^{2}$ is the phenotypic variance of the target trait [22].

### 2.6. Prediction of the Changes in Transcription Factor Binding Sites

We used JASPAR software (http://jaspar.genereg.net/, accessed on 12 February 2025) to predict the changes in the transcription factor binding sites (TFBSs) of the SNP rs109066086, which is in the 5′ UTR of the *ACOX2* gene. We inputted the sequence of each allele and its 10 bp flanking sequences separately to predict whether the transcription factor binding sites were altered (relative score ≥ 0.90).

## 3. Results

### 3.1. SNP Identification in ACOX2 Gene

We identified a total of five SNPs in the *ACOX2* gene (Figure 1), including one, rs109066086, in the 5′ UTR; rs209677248 in intron 5; and three, rs110088437, rs109665171, and rs454339362, in the 3′ flanking region. The genotypic and allelic frequencies of all the identified SNPs are summarized in Table 1.

### 3.2. Genetic Associations Between SNPs and Milk Yield and Composition Traits

By performing phenotype–genotype association analysis between the five SNPs of *ACOX2* and five milk traits (Table 2), we found that the SNP rs109066086 was significantly associated with the 305-day milk yield, milk fat yield, and milk protein yield in both the first and second lactation (*p* ≤ 4.03 × 10^−2^); rs109665171 was significantly associated with milk protein yield (*p* = 1.65 × 10^−2^); and rs454339362 was significantly associated with the 305-day milk yield, milk fat percentage, and milk protein yield in the first lactation (*p* ≤ 9.30 × 10^−3^). All five SNPs—rs109066086, rs209677248, rs110088437, rs109665171, and rs454339362—were significantly associated with the 305-day milk yield (*p* ≤ 8.00 × 10^−3^), milk fat yield (*p* ≤ 5.50 × 10^−3^), and milk protein yield (*p* ≤ 5.00 × 10^−4^) in the second lactation. Additionally, rs209677248 and rs109665171 were associated with the milk protein percentage (*p* = 3.37 × 10^−2^) and milk fat percentage (*p* = 1.17 × 10^−2^) in the second lactation, respectively. Additive, dominant, and allelic substitution effects of the five SNPs in the *ACOX2* gene were also calculated (Appendix A). We also calculated the phenotypic variance ratios (PVRs) of five SNPs for milk production traits and found that rs209677248 had higher PVRs for the 305-day milk, protein, and fat yields and milk protein percentage in the second lactation, with values of 2.94%, 2.58%, 1.83%, and 4.90%, respectively (Appendix A).

### 3.3. Genetic Associations Between Haplotypes and Milk Yield and Composition Traits

Using Haploview 4.2, we identified two haplotype blocks formed by four SNPs (D’ = 1; Figure 2). Block 1 included rs109066086 and rs209677248, in which the frequencies of H1 (CC), H2 (TC), and H3 (CG) were 0.47, 0.29, and 0.24, respectively. Block 2 included rs109665171 and rs454339362, in which the frequencies of H1 (TT), H2 (CC), and H3 (CT) were 0.33, 0.10, and 0.57, respectively. Block 1 showed significant associations with the 305-day milk, fat, and protein yields in both lactations (*p* ≤ 8.60 × 10^−3^; Table 3). Block 2 showed significant associations with the milk and protein yields in the first lactation (*p* ≤ 1.03 × 10^−2^; Table 3) and associations with the milk, fat, and protein yields and milk fat percentage in the second lactation (*p* ≤ 9.10 × 10^−4^; Table 3).

### 3.4. Transcription Factor Binding Site Changes Caused by SNPs

With JASPAR software, the allele T of the SNP rs109066086 in the 5′ UTR of the *ACOX2* gene was predicted to create a binding site for the transcription factors NR2C2 (RS = 0.91) and TFAP4 (RS = 0.90) (Table 4), implying its potential regulatory role in *ACOX2* expression. 

## 4. Discussion

Based on our previous analysis of the liver proteomes across dry and lactation stages in Holstein cows, the *ACOX2* gene was identified as a promising candidate for milk production traits. In this study, our results provided the first evidence that the *ACOX2* gene indeed has significant genetic influences on milk yield and composition traits in dairy cattle.

In human breast cancer, the expression of the *ACOX2* gene was significantly decreased (log2 ^(fold change)^ = −2.21, *p* = 6.34 × 10^−14^) [18]. Previous research has shown that the upregulation of ACOX2 can increase the production of peroxisomes and reactive oxygen species (ROS), thereby facilitating the elimination of Staphylococcus aureus in bovine mammary epithelial cells [23]. The *ACOX2* gene has been associated with physiological changes in subcutaneous and visceral fat in beef cattle [19]. A previous study has shown that the expression of *ACOX2* is downregulated in the jejunums of broilers suffering from necrotic enteritis, and this alteration may lead to disturbances in lipid metabolism, thereby compromising their immune response [24]. The above studies suggest that *ACOX2* has a critical role in the maintenance of body homeostasis and lipid metabolism.

In mammalian liver tissue, ACOX2 plays a critical role in lipid metabolism. *ACOX2* is the rate-limiting enzyme in the β-oxidation of branched-chain fatty acids, catalyzing the oxidation of long-chain fatty acids to short-chain fatty acids [16,25]. ACOX2 can also convert cholesterol to bile acids (BAs), which activate the farnesoid X receptor (FXR). The activation of FXR, in turn, induces the expression of phosphoenolpyruvate carboxykinase (PEPCK) and regulates glucose levels in human and rat hepatocytes, as well as in mouse liver [26]. In the gut microbiota, BAs are essential for the absorption, transport, and metabolism of dietary fats and fat-soluble vitamins [27]. These studies highlight the important role of *ACOX2* in lipid regulation. This is consistent with the result of the present study that *ACOX2* significantly affects milk fat traits. In addition, it was significantly associated with milk yield and milk protein traits in this study. This is probably due to the strong genetic correlation between milk traits.

SNPs can cause phenotypic variation by affecting gene function or expression [28,29]. In this study, the SNP rs109066086 in the 5′ UTR of *ACOX2* changed a predicted TFBS for the transcript factors NR2C2 and TFAP4. NR2C2 was reported to bind the promoters of 9 downregulated lncRNAs in olaparib-treated human uveal melanoma cells [30]. TFAP4 can suppress the transcription of the human homolog of murine double minute 2 (*HDM2*) gene in HCT116 cells [31,32]. For rs109066086, the TT genotype exhibited significantly lower 305-day milk, fat, and protein yields compared with the CC genotype. This may be due to the allele T producing TFBSs for the transcript factors NR2C2 and TFPA4, thereby impacting *ACOX2* expression. Thus, the SNP rs109066086 could be a potential key mutation affecting milk fat traits and deserves further in-depth study. 

In addition, we found that the H3H3 haplotype combination from block 1 was associated with higher milk, fat, and protein yields. It is possible to consider adding these SNP combinations into the chip array for a genomic selection program in dairy cattle. In the future, the dual-luciferase report assay (DLRA), chromatin immunoprecipitation (ChIP), electrophoretic mobility shift assay (EMSA), single base mutation, and other tests can be used to verify how these mutations regulate the biological functions of the *ACOX2* genes and their influence on milk production, thus providing valuable molecular markers for the genomic selection of dairy cattle.

## 5. Conclusions

In our previous proteomic research on the livers of Holstein cows, the *ACOX2* gene was identified as a potential candidate gene associated with milk production traits. This study identified polymorphisms in the *ACOX2* gene and provided the first evidence that the *ACOX2* gene had significant genetic effects on the milk yield and composition traits in dairy cattle. The SNP rs109066086 in the 5′ UTR was predicted to altered the binding sites of the transcription factors NR2C2 and TFAP4, suggesting that they might regulate *ACOX2* expression. Our findings provided valuable genetic markers for a genomic selection program in dairy cattle breeding.

## Figures and Tables

**Figure 1 animals-15-00953-f001:**
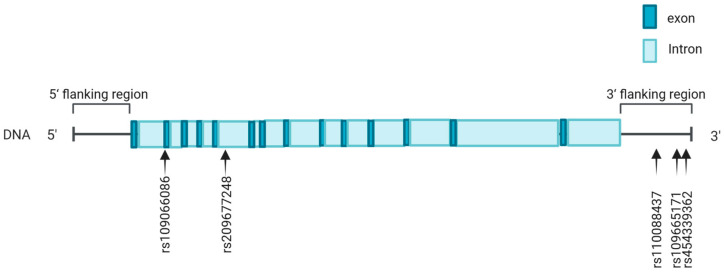
Location of SNPs in *ACOX2* gene. Dark blue boxes represent exons; light blue boxes represent introns. The arrows indicate the SNP positions.

**Figure 2 animals-15-00953-f002:**
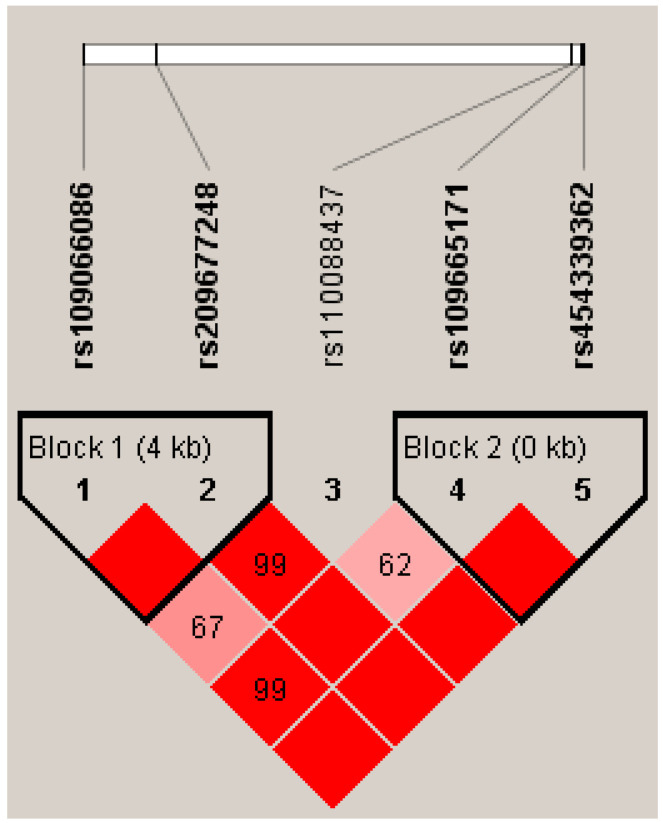
Linkage disequilibrium estimated between SNPs in *ACOX2* gene. The block indicates the haplotype block, and the text above the horizontal numbers is the SNP names. The values in boxes are pairwise SNP correlations (D′), while bright red boxes indicate complete LD (D′ = 1).

**Table 1 animals-15-00953-t001:** Detailed information about SNPs identified in the *ACOX2* gene and their genotypic and allelic frequencies.

Gene Region	SNPs	Genotype	Genotypic Frequency	Allele	Allelic Frequency
5′ UTR	rs109066086	CC	0.51	C	0.71
		TC	0.40	T	0.29
		TT	0.09		
Intron	rs209677248	CC	0.56	C	0.76
		CG	0.40	G	0.24
		GG	0.04		
3′ flanking region	rs110088437	TT	0.26	T	0.52
		TC	0.52	C	0.48
		CC	0.22		
	rs109665171	CC	0.43	C	0.67
		TC	0.47	T	0.33
		TT	0.10		
	rs454339362	TT	0.82	T	0.90
		TC	0.17	C	0.10
		CC	0.01		

**Table 2 animals-15-00953-t002:** Associations of five SNPs of the *ACOX2* gene with milk yield and composition traits in Chinese Holstein cattle during two lactations (LSM ± SE).

SNP	Lactation	Genotype (No.)	Milk Yield (kg)	Fat Yield (kg)	Fat Percentage (%)	Protein Yield (kg)	Protein Percentage (%)
rs109066086	1	CC (470)	10,322 ± 64.60 ^Aa^	341.81 ± 2.88 ^a^	3.34 ± 0.03	306.73 ± 2.10 ^Aa^	2.98 ± 0.02
		TC (367)	10,239 ± 66.11 ^Aa^	342.02 ± 2.92 ^a^	3.37 ± 0.03	304.42 ± 2.13 ^Aa^	2.99 ± 0.02
		TT (85)	10,004 ± 98.15 ^B^	333.07 ± 4.13 ^b^	3.35 ± 0.04	296.24 ± 3.00 ^B^	2.98 ± 0.03
		*p*	2.60 × 10^−3^	4.03 × 10^−2^	0.42	6.00 × 10^−4^	0.88
	2	CC (347)	9830.93 ± 68.73 ^Aa^	342.61 ± 3.07 ^Aa^	3.56 ± 0.03	284.34 ± 2.23 ^Aa^	2.97 ± 0.02
		TC (270)	9722.67 ± 71.23 ^Aa^	339.07 ± 3.14 ^Aa^	3.60 ± 0.03	280.55 ± 2.29 ^Aa^	2.99 ± 0.02
		TT (64)	9231.82 ± 114.87 ^B^	322.55 ± 4.82 ^B^	3.64 ± 0.05	263.76 ± 3.52 ^B^	2.98 ± 0.03
		*p*	<1.00 × 10^−4^	<1.00 × 10^−4^	0.16	<1.00 × 10^−4^	0.49
rs209677248	1	CC (520)	10,226 ± 62.51	339.48 ± 2.79	3.35 ± 0.03	304.21 ± 2.03	2.99 ± 0.02
		CG (366)	10,292 ± 65.91	342.72 ± 2.92	3.35 ± 0.03	305.17 ± 2.12	2.98 ± 0.02
		GG (36)	10,423 ± 135.32	349.55 ± 5.58	3.38 ± 0.05	310.03 ± 4.07	2.99 ± 0.03
		*p*	0.19	7.43 × 10^−2^	0.80	0.29	0.57
	2	CC (378)	9615.03 ± 67.60 ^Bb^	332.53 ± 3.02 ^C^	3.58 ± 0.03	277.13 ± 2.20 ^C^	2.99 ± 0.02 ^Aa^
		CG (276)	9751.25 ± 70.30 ^Ba^	342.15 ± 3.11 ^B^	3.59 ± 0.03	281.93 ± 2.27 ^B^	2.98 ± 0.02 ^ABa^
		GG (27)	10,930 ± 165.42 ^A^	389.53 ± 6.79 ^A^	3.56 ± 0.07	315.14 ± 4.95 ^A^	2.88 ± 0.04 ^Bb^
		*p*	<1.00 × 10^−4^	<1.00 × 10^−4^	0.77	<1.00 × 10^−4^	3.37 × 10^−2^
rs110088437	1	TT (239)	10,192 ± 71.72 ^b^	338.70 ± 3.13	3.36 ± 0.03	303.04 ± 2.28	2.99 ± 0.02
		TC (477)	10,253 ± 62.91 ^ab^	341.87 ± 2.81	3.36 ± 0.03	304.98 ± 2.04	2.99 ± 0.02
		CC (206)	10,363 ± 75.61 ^a^	342.12 ± 3.28	3.32 ± 0.03	306.55 ± 2.39	2.97 ± 0.02
		*p*	5.82 × 10^−2^	0.37	0.39	0.26	0.48
	2	TT (171)	9415.33 ± 80.37 ^C^	328.31 ± 3.49 ^C^	3.59 ± 0.03	270.06 ± 2.54 ^C^	2.99 ± 0.02
		TC (361)	9695.53 ± 66.54 ^B^	336.34 ± 2.97 ^B^	3.58 ± 0.03	279.55 ± 2.16 ^B^	2.98 ± 0.02
		CC (149)	10,184 ± 83.28 ^A^	359.39 ± 3.60 ^A^	3.58 ± 0.03	296.75 ± 2.62 ^A^	2.96 ± 0.02
		*p*	<1.00 × 10^−4^	<1.00 × 10^−4^	0.88	<1.00 × 10^−4^	0.53
rs109665171	1	CC (396)	10,210 ± 65.14 ^b^	340.44 ± 340.44	3.36 ± 0.03	302.83 ± 2.10 ^Bb^	2.98 ± 0.02
		TC (436)	10,279 ± 64.14 ^ab^	341.29 ± 341.29	3.35 ± 0.03	305.88 ± 2.08 ^ab^	2.99 ± 0.02
		TT (90)	10,415 ± 97.54 ^a^	343.56 ± 343.56	3.33 ± 0.04	309.78 ± 3.00 ^Aa^	2.99 ± 0.03
		*p*	6.14 × 10^−2^	0.68	0.74	1.65 × 10^−2^	0.73
	2	CC (292)	9677.97 ± 70.60 ^B^	341.78 ± 3.12 ^Aa^	3.62 ± 0.03 ^Aa^	279.88 ± 2.27 ^Aa^	2.98 ± 0.02
		TC (322)	9864.84 ± 68.12 ^A^	340.04 ± 3.04 ^Aa^	3.55 ± 0.03 ^Bb^	284.41 ± 2.21 ^Ab^	2.97 ± 0.02
		TT (67)	9160.17 ± 110.68 ^C^	320.03 ± 4.66 ^B^	3.58 ± 0.04 ^ab^	263.55 ± 3.40 ^B^	3.00 ± 0.03
		*p*	<1.00 × 10^−4^	<1.00 × 10^−4^	1.17 × 10^−2^	<1.00 × 10^−4^	0.42
rs454339362	1	TT (754)	10,304 ± 59.87 ^Aa^	341.1 ± 2.70	3.34 ± 0.02 ^Bb^	306.04 ± 1.96 ^Aa^	2.98 ± 0.02
		TC (159)	10,034 ± 81.43 ^Bb^	340.6 ± 3.50	3.42 ± 0.03 ^Aa^	298.66 ± 2.55 ^Bb^	3.00 ± 0.02
		CC (9)	10,328 ± 245.37 ^ab^	345.68 ± 9.94	3.33 ± 0.10 ^ab^	304.25 ± 7.25 ^ab^	2.96 ± 0.06
		*p*	4.00 × 10^−4^	0.87	9.30 × 10^−3^	1.30 × 10^−3^	0.62
	2	TT (558)	9776.45 ± 62.12 ^Aa^	341.2 ± 2.82 ^Aa^	3.58 ± 0.03	282.46 ± 2.05 ^Aa^	2.97 ± 0.02
		TC (115)	9520.22 ± 92.82 ^Bb^	330.42 ± 3.97 ^Bb^	3.60 ± 0.038	272.99 ± 2.89 ^Bb^	2.98 ± 0.03
		CC (8)	9658.28 ± 270.74 ^ab^	335.08 ± 10.99 ^ab^	3.57 ± 0.11	284.62 ± 8.02 ^ab^	3.04 ± 0.07
		*p*	8.00 × 10^−3^	5.50 × 10^−3^	0.79	5.00 × 10^−4^	0.60

Note: The number in the table represents the least squares mean ± standard deviation. ^a^, ^b^ within the same column with different superscripts means *p* < 0.05; ^A^, ^B^, ^C^ within the same column with different superscripts means *p* < 0.01.

**Table 3 animals-15-00953-t003:** Haplotypes analysis for blocks of the *ACOX2* gene (LSM ± SE).

Block	Lactation	Haplotype Combination (No.)	Milk Yield (kg)	Fat Yield (kg)	Fat Percentage (%)	Protein Yield (kg)	Protein Percentage (%)
Block 1	1	H1H1 (195)	10,262 ± 76.92 ^AaBb^	337.48 ± 3.33 ^ABb^	3.32 ± 0.03	305.66 ± 2.43 ^Aab^	2.99 ± 0.02
		H1H2 (240)	10,279 ± 72.28 ^Aab^	343.53 ± 3.16 ^Aa^	3.37 ± 0.03	306.08 ± 2.30 ^Aab^	2.99 ± 0.02
		H1H3 (239)	10,361 ± 73.07 ^Aa^	344.49 ± 3.19 ^Aa^	3.35 ± 0.03	307.36 ± 2.32 ^Aa^	2.98 ± 0.02
		H2H2 (85)	10,011 ± 98.21 ^Bc^	333.51 ± 4.13 ^Bb^	3.35 ± 0.04	296.43 ± 3.01 ^Bc^	2.98 ± 0.03
		H2H3 (127)	10,179 ± 84.35 ^ABbc^	340.18 ± 3.60 ^ab^	3.36 ± 0.03	301.76 ± 2.62 ^ABbc^	2.98 ± 0.02
		H3H3 (36)	10,429 ± 135.36 ^Aab^	349.71 ± 5.59 ^Aa^	3.38 ± 0.05	310.28 ± 4.07 ^Aa^	2.99 ± 0.03
		*p*	6.90 × 10^−3^	8.60 × 10^−3^	0.65	1.40 × 10^−3^	0.89
	2	H1H1 (139)	9568.54 ± 86.65 ^bC^	328.45 ± 3.73 ^CcDd^	3.54 ± 0.04 ^b^	276.60 ± 2.72 ^bC^	2.99 ± 0.02 ^ABb^
		H1H2 (175)	9784.78 ± 79.70 ^aBC^	339.57 ± 3.46 ^aBb^	3.58 ± 0.03 ^ab^	282.20 ± 2.52 ^aBCc^	2.98 ± 0.02 ^ABb^
		H1H3 (181)	9856.28 ± 80.13 ^aB^	345.62 ± 3.50 ^aB^	3.58 ± 0.03 ^ab^	285.28 ± 2.55 ^Bc^	2.97 ± 0.02 ^ABb^
		H2H2 (64)	9210.82 ± 115.00 ^D^	321.36 ± 4.83 ^Dd^	3.64 ± 0.05 ^a^	263.15 ± 3.52 ^D^	2.99 ± 0.03 ^ABb^
		H2H3 (95)	9587.56 ± 94.51 ^bC^	336.91 ± 4.00 ^BbCc^	3.62 ± 0.04 ^ab^	276.83 ± 2.92 ^abC^	2.99 ± 0.03 ^Ab^
		H3H3 (27)	10,925 ± 165.48 ^A^	389.24 ± 6.80 ^A^	3.56 ± 0.07 ^ab^	315.08 ± 4.96 ^A^	2.88 ± 0.04 ^Ba^
		*p*	<1.00 × 10^−4^	<1.00 × 10^−4^	0.38	<1.00 × 10^−4^	0.16
Block 2	1	H1H1 (90)	10,553 ± 95.65 ^Aa^	348.40 ± 4.04	3.33 ± 0.04 ^ab^	315.57 ± 2.94 ^Aa^	2.99 ± 0.03
		H1H3 (384)	10,416 ± 64.06 ^Aab^	345.31 ± 2.84	3.33 ± 0.03 ^b^	310.34 ± 2.07 ^ABb^	2.98 ± 0.02
		H2H2 (9)	10,429 ± 246.10 ^ac^	343.57 ± 9.97	3.29 ± 0.10 ^ab^	306.92 ± 7.27 ^abc^	2.95 ± 0.06
		H2H3 (107)	10,192 ± 89.13 ^Bc^	345.23 ± 3.79	3.40 ± 0.04 ^a^	304.09 ± 2.76 ^Cc^	2.99 ± 0.02
		H3H3 (280)	10,344 ± 69.79 ^ABbc^	342.45 ± 3.06	3.33 ± 0.03 ^b^	307.59 ± 2.23 ^BbCc^	2.98 ± 0.02
		*p*	1.03 × 10^−2^	0.54	0.22	3.20 × 10^−3^	0.92
	2	H1H1 (67)	8942.88 ± 108.87 ^C^	304.18 ± 4.57 ^Dd^	3.55 ± 0.04 ^abc^	254.06 ± 3.33 ^BC^	3.03 ± 0.03
		H1H2 (35)	9855.98 ± 140.50 ^A^	327.54 ± 5.80 ^ABbCc^	3.49 ± 0.06 ^ABc^	279.74 ± 4.23 ^A^	2.96 ± 0.04
		H1H3 (287)	9709.93 ± 70.33 ^A^	329.43 ± 3.13 ^Bb^	3.54 ± 0.03 ^Bc^	276.85 ± 2.28 ^A^	2.99 ± 0.02
		H2H3 (80)	9100.98 ± 101.99 ^BC^	316.08 ± 4.31 ^CcD^	3.63 ± 0.04 ^AaBb^	258.64 ± 3.14 ^B^	3.00 ± 0.03
		H3H3 (204)	9726.49 ± 76.24 ^A^	340.41 ± 3.33 ^Aa^	3.62 ± 0.03 ^Aa^	279.76 ± 2.43 ^A^	2.98 ± 0.02
		*p*	<1.00 × 10^−4^	<1.00 × 10^−4^	9.10 × 10^−3^	<1.00 × 10^−4^	0.29

Note: The number in the table represents the least squares mean ± standard deviation. ^a^, ^b^, ^c^, ^d^ within the same column with different superscripts means *p* < 0.05; ^A^, ^B^, ^C^, ^D^ within the same column with different superscripts means *p* < 0.01.

**Table 4 animals-15-00953-t004:** Prediction of transcription factor binding sites (TFBSs) for *ACOX2* gene.

GENE	SNP	Allele	Transcription Factor	Relative Score	Predicted Binding Site Sequence
*ACOX2*	rs109066086	C	—	—	—
		T	NR2C2	0.91	CAGGTGAT
			TFAP4	0.90	GCCAGGTGAT

Note: Relative scores represent the correlation score for each predicted binding site, with higher values indicating a greater likelihood of higher probability of binding. The SNP in the predicted binding site sequence is underlined.

## Data Availability

The original contributions presented in this study are included within the article/Appendix A. Further inquiries can be directed to the corresponding author(s).

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
