# Peer review of "Genetic Associations of ACOX2 Gene with Milk Yield and Composition Traits in Chinese Holstein Cows"

_animals, 2025, doi:10.3390/ani15070953_

Round 1

Reviewer 1 Report

Comments and Suggestions for Authors

The data provided in the study complement the existing information on the relationship of the cow genome to human and economically important milk parameters. It offers further study opportunities in many breeds and improves the understanding of the structure and function of animal genetics.

The publication is written in clear and accessible English, which, I believe, will be easily understood by specialists and experts in a particular field.

A couple of things that apply to the entire article rather than a specific chapter:

P values: If you want to specify more than two decimal places, I recommend using a statistical format, for example, 1.23 x10-3.

Given that a specific gene is being analysed, I recommend using gene localisation rather than the genome for SNP identification, especially in a gene image. Second, I would still recommend using SNP ID numbers in the abstract and text to make future searches more accurate, especially considering localisation numbering changes over time.

Title

Reflects the topic completely

Abstract:

Very good and concise“Simple summary”.

Overall, the abstract is easy to understand and shows both the beginning of the study and the most important results.

In line 33- 34 of the abstract, there are two P values, but before the brackets there are three features: “… associated with 305-day milk, fat, and protein yields (P ≤ 0.0086; P ≤ 0.0001). “

Introduction.

The introductory material is well organized and contains essential information to explain the rationale for conducting the study. It would only be good to have some additional information about the variety used and its distribution.

Materials and methods.

Very well-described methods and a good sequence were chosen to show that the study has been carried out sequentially and thoughtfully

Results

As I mentioned at the beginning of the review, two decimal places are sufficient for numerical data. For the frequencies in Table 1, two decimal places are sufficient. However, if you want more precision, I would recommend expressing the genotype and allele distributions in percentages rather than decimals.

Also, Table 1 would recommend separating the 3'UTR end SNPs with a horizontal line to separate the genotype data. At the moment, everything in the table is a bit mixed up.

Two decimal places and horizontal lines also apply to Table 2 and Table 3.

Was the r2 value also determined in the LD analysis to be able to analyse the existence of LD?

Discussion

The discussion is very short. Only one of the 5 SNPs found to be associated with milk parameters has been discussed. I recommend supplementing it with an analysis of the rest of the results:

> Have the SNPs been analysed/genotyped in other breeds/populations?

> What could the obtained associations mean?

> Do individually derived associations show up in haplotype, or more precisely, multi-locus genotype analysis?

> How could obtained results be used?

> What could be the possible functional significance of the other SNPs that were found to be associated with milk parameters

Reviewer 2 Report

Comments and Suggestions for Authors

Dear authors,
Dear Editor,

I have carefully reviewed the article entitled Genetic associations of ACOX2 gene with milk yield and compositiontriats in Chinese Holstein cows.

Overall, the article is well written, presents numerous novel elements that are based on the results obtained through a modern series of working techniques and models in genetics. Without a doubt, it is a competitive article and suitable for publication in MDPI journals.

The authors missed a few details and I think it is good to review the content of the article, starting with a few spelling mistakes but also some additions that I think are welcome.

Row 25: space betweeen intron and 6
Row 61: delete a space between of and by, and even delete of
Row 65: space before 16
Row 73: is regarde nor regared
Row 74: promising not promsing
Row 133: space between allele and B
Row 137: delete underline after the brackets

I think you can take into account that in table 2 the data should be expressed with a maximum of 3 digits after the point.

The discussion chapter puts far too little value on the entire work of the authors. Here I think that consistent work is still needed, with the introduction of comparative data from other breeds of cows, and even species, for the influence of the ACOX2 gene on milk production.

Moreover, in the discussions or perhaps in the conclusions, I think that an emphasis on the importance of the study carried out would be welcome.

Comments on the Quality of English Language

English is fine, only some spell errors that were suggested to be corrected. 

Reviewer 3 Report

Comments and Suggestions for Authors

This manuscript explores the genetic effects of the ACOX2 gene on milk production traits in Chinese Holstein cows. By sequencing the entire coding region along with the flanking sequences, the study identifies five SNPs, some of which are significantly associated with key milk production traits across lactations. Haplotype analysis further reveals two blocks significantly linked to 305-day milk, fat, and protein yields. Additionally, the study predicts that the T allele in SNP g.42838675C>T creates transcription factor binding sites (TFBSs) for NC2R2 and TFAP4, potentially influencing ACOX2 expression. These findings provide valuable insights into the genetic basis of milk traits and highlight molecular markers that may aid genomic selection in dairy cattle.

Overall, the manuscript is well written and presents an interesting and useful topic. My comments focus primarily on minor issues related to consistency, formatting, and clarity.

GENERAL COMMENTS

  1. I recommend including RS IDs alongside genomic coordinates (e.g., g.42838675C>T) when reporting SNPs. RS IDs provide a stable and widely recognized reference that does not change with genome build updates, making the study more accessible to researchers. Including them can also enhance searchability and visibility, as RS IDs are commonly used in databases like dbSNP, Ensembl, and GWAS Catalog. Using RS IDs can improve the study’s visibility and impact, making it easier for researchers to link their findings to your results, potentially increasing citations.
  1. When referring to genes, use italics (ACOX2); when referring to proteins, use regular font (ACOX2). For example:
    1. Line 60: Refers to a proteomic study, so it should be ACOX2 (not ACOX2).
    2. Lines 209, 211, 217: Refer to the enzyme, so they should be ACOX2 (not ACOX2).
    3. Line 243: Refers to the gene, so it should be ACOX2 (not ACOX2).
  1. Throughout the manuscript: Consider using "altered predicted TFBS" instead of "altered TFBS" to reflect that the results are based on computational predictions from JASPAR rather than direct experimental validation.

GENERAL TEXT REVIEW – FORMATING AND CLARITY

Below are some inconsistencies and errors I noticed. Please review the text carefully for similar issues throughout.

  1. Line 25 and 143: Please use "intron 6" instead of "intron6" for clarity and consistency with standard genetic notation.
  2. Line 61: "." is missing after the citation—should be: "compared to the dry period (fold change ≥ 1.5, P = 0.006749) [14]."
  3. Lines 61-64: The phrase "biosynthesis of by" is unclear—please revise for clarity.
  4. Line 65: "intermediates[16]." has no space before the citation—add a space ("intermediates [16].").
  5. Lines 68-70: The phrase "with 1.8 Mb, and the SNP..." is unclear. Please clarify whether "with 1.8 Mb" and "with 0.2 Mb" refer to the distance from the ACOX2 Rewording would improve clarity and readability.
  6. Line 225: Extra ".[28]." before the citation—remove redundant punctuation.
  7. Please ensure that decimal places are consistent within each column in tables. Some values are reported with two decimal places (10262.00, 305.66), while others have three (3.320, 2.992). Please ensure consistency in decimal places within each table column. For example, large whole numbers may not need two decimal places (e.g., 10262.00 → 10262), while smaller values should have uniform precision (e.g., 3.320 and 2.992 should match). Additionally, use a logical number of decimal places throughout the manuscript. For example, in Line 61 (fold change ≥ 1.5, P = 0.006749), excessive decimal places are unnecessary, and rounding to P = 0.007 would be clearer.
  8. Tables 2 and 3: In the notes below the tables, "P<0,5" should include spaces for proper formatting: "P < 0.5" instead of "P<0,5". Please ensure consistent spacing in statistical expressions throughout the manuscript.

ABSTRACT

  1. Line 35: Replace "transcript factors" with "transcription factors".

MATERIAL AND METHODS

  1. Line 95: The authors must describe why the 2000 bp 5' and 3' flanking sequences are suitable.
  2. Section 2.6: Please describe how JASPAR was used to analyze allele-specific TF binding. Did you input sequences separately for each allele to compare binding scores? Clarifying this would improve reproducibility.

Reviewer 4 Report

Comments and Suggestions for Authors

Simple summary: well written, no comments

Abstract: authors did a good job here so nothing to comment

Material and methods: Why authors used semen as source of bull DNA not blood? Is there any reason?

Data analysis part looked clear and okay to me but I am not an expert here.

Results: Are you sure location of SNP’s in flanking region did any significant change?

Then again, your results were stated very clearly in these big tables so nothing to say here.

Discussion: no comments but can be bit elaborated comparing and contrasting with other published data regarding this same gene, if available

Conclusion: this is a good follow up study, and authors mentioned that properly there.
